# Upregulation of Mineralocorticoid Receptor Contributes to Development of Salt-Sensitive Hypertension after Ischemia–Reperfusion Injury in Rats

**DOI:** 10.3390/ijms23147831

**Published:** 2022-07-15

**Authors:** Takumi Matsumoto, Shigehiro Doi, Ayumu Nakashima, Takeshi Ike, Kensuke Sasaki, Takao Masaki

**Affiliations:** 1Department of Nephrology, Hiroshima University Hospital, 1-2-3 Kasumi, Minami-ku, Hiroshima 734-8551, Japan; takumi.huniver@gmail.com (T.M.); ayumu@hiroshima-u.ac.jp (A.N.); takeshieki@gmail.com (T.I.); kensasaki@hiroshima-u.ac.jp (K.S.); 2Doi Nephrology & Dialysis Clinic, 5-18-6 Midori, Minami-ku, Hiroshima 734-0005, Japan; 3Department of Stem Cell Biology and Medicine, Graduate School of Biomedical & Health Sciences, Hiroshima University, 1-2-3 Kasumi, Minami-ku, Hiroshima 734-8553, Japan

**Keywords:** mineralocorticoid receptor, acute kidney injury, salt-sensitive hypertension, epithelial sodium channel

## Abstract

The ischemia–reperfusion injury (IRI) of rat kidneys is used as a model of acute kidney injury. Salt-sensitive hypertension occurs in rats after IRI, and the distal nephrons play important roles in the development of this condition. We investigated the role of the mineralocorticoid receptor (MR) in the progression of IRI-induced salt-sensitive hypertension in rats. Fourteen days after right-side nephrectomy, IRI was induced by clamping the left renal artery, with sham surgery performed as a control. IRI rats were provided with normal water or water with 1.0% NaCl (IRI/NaCl), or they were implanted with an osmotic mini-pump to infuse vehicle or aldosterone (IRI/Aldo). Esaxerenone, a non-steroidal MR blocker (MRB), was administered to IRI/NaCl and IRI/Aldo rats for 6 weeks. MR expression increased by day 7 post-IRI. Blood pressure and urinary protein excretion increased in IRI/NaCl and IRI/Aldo rats over the 6-week period, but these effects were negated by MRB administration. The MRB attenuated the expression of the gamma-epithelial sodium channel (ENaC) and renal damage. The ENaC inhibitor, amiloride, ameliorated hypertension and renal damage in IRI/NaCl and IRI/Aldo rats. Our findings thus showed that MR upregulation may play a pivotal role in ENaC-mediated sodium uptake in rats after IRI, resulting in the development of salt-sensitive hypertension in response to salt overload or the activation of the renin–angiotensin–aldosterone system.

## 1. Introduction

Chronic kidney disease (CKD) is defined as the presence of albuminuria and/or decline in the estimated glomerular filtration rate and affects approximately 700 million individuals worldwide [1,2,3,4]. CKD is, therefore, currently well-recognized as a world health concern. In a clinical setting, patients with CKD eventually develop end-stage kidney disease, which requires renal replacement therapy or transplantation [5]. Although various diseases can lead to CKD, it is particularly frequent in patients with a history of acute kidney injury (AKI) [6,7], and the AKI-to-CKD transition is, thus, a well-recognized major pathway for progression to CKD [8,9,10]. However, the mechanisms by which AKI causes CKD remain unclear.

Hypertension participates in the development of cardiovascular diseases, such as coronary artery disease, stroke, and peripheral artery disease [11,12,13]. CKD is also currently recognized as a vascular disease [14], and hypertension contributes to the promotion of renal dysfunction regardless of the primary disease [15,16,17,18]. Notably, hypertension frequently occurs in patients with a history of AKI, even if their serum creatinine levels have returned to within the normal range [19]. The ischemia–reperfusion injury (IRI) of the kidney is used as a rodent model of AKI, and salt overload in IRI rats causes an increase in their blood pressure in IRI rats [20,21]. Different from essential hypertension, hypertension in response to salt overload is termed “salt-sensitive hypertension”, which contributes to renal damage [22]. These findings imply that increased salt uptake occurs post-AKI, which may participate in the AKI-to-CKD transition.

The mineralocorticoid receptor (MR) is expressed in the kidney and various other organs, including the heart, colon, and brain [23]. Although the MR binds to deoxycorticosterone and glucocorticoids, aldosterone is the main MR ligand, and it activates the renin–angiotensin–aldosterone system (RAAS) [24,25]. In the kidneys, the MR functions to promote sodium reabsorption by the distal nephron through the epithelial sodium channel (ENaC) and the thiazide-sensitive sodium-chloride cotransporter (NCC) [26,27]. MR expression was reported to be increased in astrocytes and microglia in the hippocampal CA1 region after brain IRI [28]. These findings suggest that MR may be upregulated after renal IRI and may have an important role in the development of hypertension.

Experimentally, salt-sensitive hypertension can be induced in Sprague Dawley rats by both salt overload and the activation of the RAAS [29]; however, as mentioned above, salt overload alone causes hypertension in IRI rats. We hypothesized that the IRI-induced upregulation of MR in the distal nephron contributes to the increased expression of ENaC and NCC, allowing salt overload to cause hypertension without the need for the activation of the RAAS. Moreover, we hypothesized that the upregulation of the MR increases reactivity to aldosterone and that the administration of aldosterone alone leads to hypertension in the absence of salt overload. To confirm these hypotheses, we investigated whether salt overload or aldosterone infusion caused hypertension in IRI rats and determined the functional roles of MRs and sodium channels in the distal nephron.

## 2. Results

### 2.1. MR Expression Increases in IRI Rats at Day 7 Post-Surgery

Although MR expression is upregulated after brain IRI [28], the effects of renal IRI on MR expression and its downstream effectors in the distal nephron remain unknown. We evaluated the expression levels of MR, serum, and glucocorticoid-regulated kinase 1 (SGK1), ENaC, and NCC in the kidneys of IRI rats by Western blotting on day 7 after IRI. MR and SGK1 expression levels increased in IRI rats, compared with the levels in sham rats on day 7 (Figure 1A,B), and increased MR expression persisted at the end of the sixth week (Appendix A). The expression levels of β- and γ-ENaC increased significantly in IRI rats, but there was no significant difference in the expression of α-ENaC between IRI and sham rats. Previous studies reported that ENaC was activated after processing full-length γ-ENaC (83 kDa) to its cleaved, active 70 kDa form [30,31,32]. In this study, both full-length and cleaved γ-ENaC increased in IRI rats, while the expression of phospho-NCC (Thr53) (p-NCC^T53^) did not differ between IRI and sham rats (Figure 1A,B). There was no significant difference in plasma aldosterone concentrations between sham and IRI rats 7 days after sham or IRI operation (Figure 1C).

### 2.2. NaCl Overload Causes Hypertension and Renal Damage in IRI Rats

To investigate the effect of salt overload alone in a rat model of AKI, we compared blood pressure and renal damage in IRI rats provided with water or 1.0% NaCl solution (IRI/NaCl). Systolic blood pressure increased significantly from the second week post-surgery in the IRI/NaCl group, compared with that in the other groups (Figure 2A). The urinary excretion of albumin, Na^+^, and Cl^−^ increased in the IRI/NaCl group, compared with levels in the IRI rats, but K^+^ excretion did not differ between the IRI/NaCl and IRI groups (Figure 2B). Renal damage was assessed by the histological analysis of renal tissue. Hematoxylin–eosin (HE) staining revealed that renal tissue from IRI/NaCl rats had increased cellularity and tubular dilation compared with that from IRI rats, while Masson’s trichrome (MT) staining showed that renal fibrosis (positive area with aniline blue) was enhanced in IRI/NaCl rats. Immunohistochemical staining demonstrated that the expression of extracellular matrix marker collagen type 3 (Col-III) was increased in tissues from IRI/NaCl rats, compared with that in tissues from the other three groups (Figure 2C). Western blotting revealed that alpha-smooth muscle actin (α-SMA) expression, as a marker of myofibroblasts, was upregulated in IRI/NaCl rats, compared with that in the IRI and sham groups (Figure 2D). However, there was no significant difference in plasma aldosterone concentrations between the IRI and IRI/NaCl rats (n = 5 per group) (Appendix A).

### 2.3. Aldosterone Infusion Induces Hypertension and Renal Damage in IRI Rats

We assessed the effect of aldosterone administration alone in the rat model of AKI by examining blood pressure and renal damage in IRI rats infused with vehicle or aldosterone (IRI/Aldo). Systolic blood pressure gradually increased in IRI/Aldo rats despite the absence of salt overload, with a significant difference beginning in the second week after surgery compared with systolic blood pressure in the IRI and sham groups (Figure 3A). Urinary albumin excretion increased in IRI/Aldo rats, but urinary electrolyte excretion did not differ among the four groups (Figure 3B). We performed HE and MT staining to assess renal tissue damage (Figure 3C). Cellularity increased in the tubulointerstitium (HE stain) and fibrosis increased in the kidneys (MT stain) of IRI/Aldo rats, compared with the levels in IRI rats. Immunohistochemistry staining demonstrated that the expression of Col-III was upregulated in the kidneys of IRI/Aldo rats (Figure 3C), and Western blotting revealed that the expression of α-SMA was enhanced in IRI/Aldo rats, compared with the levels in IRI rats (Figure 3D).

### 2.4. MRB Ameliorates Hypertension and Renal Damage in IRI/NaCl Rats

MR expression was upregulated in IRI rats, and salt overload alone induced hypertension and renal damage in these rats. We then examined whether the non-steroidal MR blocker (MRB) esaxerenone ameliorated hypertension and renal damage in IRI/NaCl rats. All rats were provided with water containing 1.0% NaCl after sham or IRI operation and administrated with vehicle or MRB. The MRB suppressed the rise in blood pressure and increased urinary albumin excretion (Figure 4A,B) and significantly ameliorated the changes in renal tissue cellularity and fibrosis in IRI/NaCl rats (Figure 4C). We also performed immunohistochemical staining to evaluate the infiltration of inflammatory cells. Esaxerenone suppressed the expression levels of CD3 and CD68, as markers of T cells and macrophages, respectively, in renal tissues from IRI/NaCl rats (Appendix A). Western blotting also showed that esaxerenone suppressed α-SMA expression in IRI/NaCl rats (Figure 4D). Expression levels of MR, β-ENaC, full-length γ-ENaC, and cleaved γ-ENaC were all increased in IRI/NaCl rats, compared with those in sham/NaCl rats, and the expression levels of these proteins were decreased by esaxerenone administration. However, the expression levels of α-ENaC and p-NCC^T53^ did not differ between IRI/NaCl and IRI/NaCl + MRB rats (Figure 4D).

### 2.5. MRB Ameliorates Hypertension and Renal Damage in IRI/Aldo Rats

We demonstrated that MR expression was upregulated in IRI rats and that the administration of aldosterone alone caused hypertension and renal damage after IRI. We, therefore, examined whether esaxerenone ameliorated hypertension and renal tissue damage in IRI/Aldo rats. All rats were infused with aldosterone after sham or IRI operation and administrated with vehicle or MRB. The rise in blood pressure and excretion of urinary albumin observed in the IRI/Aldo group was suppressed in the IRI/Aldo + MRB group (Figure 5A,B). Furthermore, the changes in cellularity, fibrosis, and inflammation in IRI/Aldo rats were ameliorated in the kidneys of IRI/Aldo +MRB rats (Figure 5C, Appendix A). Esaxerenone decreased the expression levels of α-SMA, MR, full-length γ-ENaC, and cleaved γ-ENaC induced in IRI/Aldo-treated rats. However, the expression levels of α-ENaC, β-ENaC, and p-NCC^T53^ did not differ between IRI/Aldo and IRI/Aldo + MRB rats (Figure 5D).

### 2.6. AML Ameliorates Hypertension and Renal Damage in IRI/NaCl Rats

We showed that esaxerenone ameliorated hypertension and renal damage in IRI/NaCl rats, accompanied by the downregulation of β-ENaC, full-length γ-ENaC, and cleaved γ-ENaC. We, therefore, investigated whether the oral administration of amiloride (AML), as an ENaC inhibitor, ameliorated hypertension and renal damage in IRI/NaCl rats. All rats were provided with water containing 1.0% NaCl after sham or IRI operation and administrated with vehicle or AML. The results were similar to those described above. Blood pressure and the urinary excretion of albumin were suppressed, and tissue cellularity, fibrosis, and inflammation were improved in IRI/NaCl + AML rats, compared with the levels in the IRI/NaCl group (Figure 6A–C, Appendix A). The expression levels of α-SMA, β-ENaC, full-length γ-ENaC, and cleaved γ-ENaC, but not MR, were decreased in IRI/NaCl rats by AML treatment, while the levels of α-ENaC were similar in all four groups (Figure 6D).

### 2.7. AML Ameliorates Hypertension and Renal Damage in IRI/Aldo Rats

We demonstrated that esaxerenone treatment ameliorated hypertension and renal damage in IRI/Aldo rats and downregulated full-length and cleaved γ-ENaC proteins. We, therefore, investigated whether the oral administration of AML ameliorated blood pressure and renal damage in IRI/Aldo rats. All rats were infused with aldosterone after sham or IRI operation and administrated with vehicle or AML. Consistent with the above experiments, the blood pressure rise and albumin excretion were suppressed, and tissue cellularity, fibrosis, and inflammation were improved in the IRI/Aldo + AML group, compared with levels in the IRI/Aldo group (Figure 7A–C, Appendix A). Western blotting revealed that expression levels of α-SMA, β- ENaC, full-length γ-ENaC, and cleaved γ-ENaC, but not MR expression, were decreased in IRI/Aldo + AML rats, compared with the levels in IRI/Aldo rats, while the expression of α-ENaC did not differ among the four groups (Figure 7D).

## 3. Discussion

In this study, we showed that MR expression increased in rat kidneys on day 7 after IRI, accompanied by the upregulation of SGK1 and β- and γ-ENaC but not α-ENaC or p-NCC ^T53^. Treatment with either salt overload or aldosterone caused hypertension, proteinuria, and renal damage in IRI rats, whereas the administration of esaxerenone, a non-steroidal MRB, ameliorated these and downregulated the MR and γ-ENaC. Furthermore, the administration of the ENaC inhibitor AML improved hypertension and renal damage in IRI/NaCl and IRI/Aldo rats, with a concomitant decrease in γ-ENaC expression. These findings suggested that the MR upregulation enhanced sensitivity to both salt overload and increased aldosterone levels in rats with IRI and may thus play a pivotal role in the development of hypertension and renal damage through the ENaC-mediated uptake of sodium. AKI-induced salt-sensitive hypertension may, therefore, be a therapeutic target for the AKI-to-CKD transition.

We demonstrated that MR expression increased in the kidneys of IRI rats. Past studies reported that the MR expression increased in the hippocampus of gerbils after the IRI induction and also in human subjects who were successfully resuscitated after cardiopulmonary arrest [28,33], indicating that the MR expression increases in tissues after IRI. The production of epidermal growth factor (EGF) was enhanced in rats after IRI, and increased levels of EGF resulted in the upregulation of the MR by inhibiting degradation [34]. However, although we observed increased expression levels of EGF, EGF receptor (EGFR), and phospho-EGFR in IRI rats on day 7, the EGFR inhibitor AG1478 failed to decrease the expression of MR, even with the suppression of EGFR phosphorylation (Appendix A). Therefore, the mechanism by which IRI intensifies the MR expression remains obscure. Moreover, IRI is implicated in transplantation-induced renal injury as well as AKI [35], and randomized controlled trials revealed that MRBs had beneficial effects on kidney transplant recipients [36,37]. Another randomized controlled trial reported that the MRB spironolactone improved the estimated glomerular filtration rate and proteinuria in CKD patients [38]. The IRI-induced upregulation of the MR is, therefore, considered to play a role in the AKI-to-CKD transition.

Previous studies reported that both salt overload and the activation of the RAAS were required to develop hypertension in rats [29,39]. In this study, we demonstrated that salt overload induced hypertension in rats after IRI, even without aldosterone infusion, suggesting that post-AKI patients should refrain from excessive salt intake to avoid the progression of hypertension. We also demonstrated that esaxerenone suppressed hypertension and renal damage in IRI rats during salt overload. Past studies reported that salt overload alone caused hypertension in salt-sensitive Dahl rats and rats that underwent 5/6 nephrectomy, and that the administration of an MRB ameliorated the rise in blood pressure in these animals [40,41]. The MR thus appears to play an important role in the animal models of salt-sensitive hypertension. In this study, we showed that the expression levels of β- and γ-ENaC increased in IRI rats and that the MRB esaxerenone suppressed the expression levels of these proteins. These findings suggest that the upregulation of MR promotes the activation of ENaC, and salt overload alone thus induces hypertension in rats with IRI.

Aldosterone induced hypertension in rats with IRI without salt overload in our study, and esaxerenone suppressed aldosterone-induced hypertension and renal damage. The increased expression of MR has been reported to promote sensitivity to aldosterone [42], suggesting that aldosterone infusion intensified sodium reabsorption in our IRI rats. Our findings also indicated that the expression levels of β- and γ-ENaC were increased in aldosterone-infused IRI rats, compared with the levels in sham-surgery rats infused with aldosterone. These findings suggest that ENaC-mediated sodium reabsorption is increased under the conditions of both MR upregulation and activation of RAAS, even with a normal salt diet. Clinically, the RAAS is activated in various conditions, such as chronic heart failure, obesity, and diabetes [43,44,45], and a strict low-salt diet or the use of a RAAS inhibitor may thus be required to prevent salt-sensitive hypertension in post-AKI patients with chronic heart failure, obesity, or diabetes.

The ENaC plays an important role in maintaining sodium homeostasis and body-fluid volume, but an excessive uptake of sodium contributes to the progression of salt-sensitive hypertension [46,47,48]. Additionally, the activation of the ENaC causes hypertension in some hereditary diseases, such as pseudohypoaldosteronism type 1 and Liddle syndrome [49,50,51]. Among the three components of the ENaC, γ-ENaC expression increased in IRI rats with salt overload or aldosterone infusion and was reduced by esaxerenone treatment. These results suggest that the ENaC is upregulated as a result of the increased expression of the MR. As mentioned above, the cleaved form of γ-ENaC plays a pivotal role in promoting an ENaC-mediated uptake of Na^+^ [30,31,32]. In this study, the expression levels of both full-length and cleaved γ-ENaC were increased in IRI rats with salt overload or aldosterone infusion and were suppressed by the administration of an MRB or AML. These findings suggest that MRB and AML reduced the expression of γ-ENaC rather than its processing in IRI rats with salt overload or aldosterone infusion.

In addition to hypertension, esaxerenone suppressed renal injury in IRI rats with salt overload or aldosterone infusion. The steroidal MRB spironolactone has been reported to prevent renal fibrosis and inflammation after IRI [52,53,54]. These findings raise the possibility that the inhibition of the MR ameliorates IRI-induced renal damage and thereby suppresses hypertension. However, we also demonstrated that AML improved hypertension and renal damage in IRI rats with salt overload or aldosterone infusion. The ENaC-mediated uptake of Na^+^ may, therefore, precede the development of renal damage. Taken together, these results indicate that esaxerenone protects against renal damage by inhibiting Na^+^ uptake in IRI rats with salt overload or aldosterone infusion.

## 4. Materials and Methods

### 4.1. Animals

Six-week-old male Sprague Dawley rats were purchased from Charles River Laboratories Japan (Yokohama, Japan) and fed with standard rat chow (0.48% NaCl; Oriental Yeast Co., Tokyo, Japan). Rats underwent right-sided nephrectomy or sham surgery after anesthesia via intraperitoneal injection of medetomidine–midazolam–butorphanol [55]. Fourteen days after nephrectomy, the left renal vascular pedicle was clamped for 45 min or left unclamped (sham control) using Micro Serrefines (Fine Science Tools, Foster City, CA, USA). Aldosterone (Sigma-Aldrich; St. Louis, MO, USA) was dissolved in distilled water containing dimethyl sulfoxide, and the solution was infused into rats using subcutaneously implanted Alzet osmotic pumps (Durect, Cupertino, CA, USA) [56], which provided a constant aldosterone infusion of 0.75 µg/h throughout the 6-week study period. The non-steroidal MRB esaxerenone was kindly provided by Daiichi Sankyo Co., Ltd. (Tokyo, Japan) and administered via oral gavage. Methylcellulose (0.5 *w*/*v* %) was used as the vehicle. Amiloride (AML; Sigma-Aldrich) was administered via oral gavage, using distilled water as the vehicle. All experiments were conducted in accordance with the National Institutes of Health Guidelines for the Use of Laboratory Animals. The Institutional Animal Care and Use Committee of Hiroshima University (Hiroshima, Japan) approved the experimental protocols (permit number A19-68). All efforts were taken to minimize the pain and suffering of the animals.

### 4.2. Experimental Protocol

Experiment 1. The MR expression was assessed after euthanizing the rats by cardiac puncture under deep anesthesia on day 7 after sham or IRI operation (n = 5 per group).

Experiment 2. The effect of salt overload after AKI was investigated by allocating the rats into four groups 7 days after recovery from sham or IRI surgery (n = 6 per group): Group 1, distilled drinking water after sham operation (sham rats); Group 2, 1.0% NaCl in drinking water after sham operation (sham/NaCl rats); Group 3, distilled drinking water after IRI operation (IRI rats); and Group 4, 1.0% NaCl in drinking water after IRI operation (IRI/NaCl rats). At the end of the sixth week, the rats were euthanized via cardiac puncture under deep anesthesia.

Experiment 3. The effect of aldosterone treatment in AKI rats was assessed by dividing the rats into four groups 7 days after sham or IRI operation (n = 6 per group): Group 1, vehicle infusion after sham operation (sham rats); Group 2, aldosterone infusion after sham operation (sham/Aldo rats); Group 3, vehicle infusion after IRI operation (IRI rats); and Group 4, aldosterone infusion after IRI operation (IRI/Aldo rats). At the end of the sixth week, the rats were euthanized via cardiac puncture under deep anesthesia.

Experiment 4. To determine if physiological and histological changes in IRI/NaCl or IRI/Aldo rats could be ameliorated by MRB, all rats were provided with water containing 1.0% NaCl or infused with aldosterone 7 days after recovery from sham or IRI operation and allocated into four groups (n = 6 per group): Group 1, 0.5 *w*/*v* % methylcellulose administration after sham operation (sham/NaCl or sham/Aldo rats); Group 2, 3 mg/kg/day esaxerenone administration after sham operation (sham/NaCl + MRB or sham/Aldo + MRB rats); Group 3, methylcellulose administration after IRI operation (IRI/NaCl or IRI/Aldo rats); and Group 4, esaxerenone administration after IRI operation (IRI/NaCl + MRB or IRI/Aldo + MRB rats). At the end of the sixth week, the rats were euthanized via cardiac puncture under deep anesthesia.

Experiment 5. To assess if physiological and histological changes in IRI/NaCl or IRI/Aldo rats could be ameliorated by AML, all rats were provided with 1.0% NaCl in their drinking water or infused with aldosterone on day 7 after recovery from sham or IRI operation and allocated into four groups (n = 6 per group): Group 1, distilled water administration after sham operation (sham/NaCl or sham/Aldo rats); Group 2, 5 mg/kg/day AML administration after sham operation (sham/NaCl + AML or sham/Aldo + AML rats); Group 3, distilled water administration after IRI operation (IRI/NaCl or IRI/Aldo rats); and Group 4, AML administration after IRI operation (IRI/NaCl + AML, IRI/Aldo + AML rats). At the end of the sixth week, the rats were euthanized via cardiac puncture under deep anesthesia.

### 4.3. Measurement of Biological Parameters

Blood pressure was measured weekly using an automatic sphygmomanometer (Softron, Tokyo, Japan). Twenty-four-hour urine samples were collected biweekly using metabolic cages (Natsume, Tokyo, Japan). At the end of the sixth week, blood samples were outsourced to Nikken Seil Co., Ltd. (Tokyo, Japan), and urine samples were outsourced to SRL, Inc. (Tokyo, Japan), for analysis.

### 4.4. Histology and Immunohistochemistry Analysis

Renal tissue samples were formalin-fixed and paraffin-embedded. The blocks were either cut into 2 µm thick sections and stained with hematoxylin–eosin (HE) or Masson’s trichrome (MT), or cut into 4 µm thick sections and subjected to immunohistochemical staining using anti-collagen type 3 (Col-III) antibody (1:1000; AB_306066; ab7778; Abcam, Cambridge, UK), anti-CD3 antibody (undiluted; AB_2732001; IS503; Dako, Glostrup, Denmark), and anti-CD68 antibody (1:1000; AB_2291300; MCA341R; BioRad, Hercules, CA, USA). Immunohistochemistry images without primary antibodies were used as negative controls (Appendix A). All microscopic images were captured using Lumina Vision 2.20 (Mitani Corp., Osaka, Japan). The areas of interstitial fibrosis observed after MT staining were assessed using Lumina Vision, and five randomly selected fields (×100) of the corticomedullary junction were examined. Positive staining for Col-III was assessed by examining the five selected fields (×100) using ImageJ software (National Institutes of Health; Bethesda, MD, USA).

### 4.5. Western Blot Analysis

To extract protein for Western blotting, frozen renal tissue samples were lysed in a Laemmli buffer (Sigma-Aldrich) or cell lysis buffer (Cell Signaling Technology, Danvers, MA, USA) and homogenized for 30 s using a VP-50 ultrasonic homogenizer (Taitec Corp., Saitama, Japan) at full power. Soluble protein was additionally sonicated three times for 30 s each. The concentrations of protein in the lysates were measured using a Pierce BCA protein assay kit (Thermo Fisher Scientific, Waltham, MA, USA) and adjusted so that equal amounts of total protein were analyzed in each sample, as described previously [57]. The following primary antibodies were used: anti-MR antibody (1:5000; AB_303287; ab2774; Abcam); anti-serum and glucocorticoid-regulated kinase 1 (SGK1) antibody (1:2500; AB_11060451; NBP1-76578; Novus Biologicals, Littleton, CO, USA); anti-epithelial sodium channel-α, β, and γ antibodies (1:5000; AB_10640131; SPC-403D, 1:2500; AB_10644173; SPC-404D, and 1:2500; AB_10640369; SPC-405D, respectively; StressMarq Biosciences, Victoria, BC, Canada); anti-phospho-thiazide-sensitive NaCl cotransporter (phospho T53) (p-NCC^T53^) antibody (1:2500; NBP2-60775, Novus Biologicals); anti-α-smooth muscle actin (α-SMA) antibody (1:5000; AB_476701; A2547; Sigma-Aldrich), and anti-glyceraldehyde-3-phosphate dehydrogenase (GAPDH) antibody (1:5000; AB_1078991; G8795; Sigma-Aldrich). The secondary antibodies were horseradish-peroxidase-conjugated goat anti-rabbit and goat anti-mouse immunoglobulins (Dako). Protein signals were detected using Super Signal West Pico (Thermo Fisher Scientific) or ImmunoStar^®^ LD (Fujifilm Wako Pure Chemical Corporation, Osaka, Japan) chemiluminescence reagents, and the intensity of each band was analyzed using ImageJ software.

### 4.6. Statistical Analysis

The data are summarized as mean ± standard error for each group of rats. Comparisons between two groups were tested using Student’s *t*-test and multiple comparisons were tested using a one-way analysis of variance with Tukey’s post hoc tests. *p* values < 0.05 were considered statistically significant.

## 5. Conclusions

This study showed that MR expression increased in IRI rats, and salt overload or aldosterone infusion alone could cause hypertension, proteinuria, renal tissue damage, and increased MR and γ-ENaC expression. Moreover, the MRB esaxerenone ameliorated these conditions, and AML also improved these conditions by suppressing γ-ENaC expression in IRI rats during salt overload or aldosterone infusion. However, the expression of p-NCC did not change in this study. These findings suggest that the upregulation of the MR promotes salt-sensitive hypertension in response to salt overload or the increased activity of the RAAS in rats with IRI, and the ENaC-mediated reabsorption of sodium may play an important role in the development of salt-sensitive hypertension in rats after IRI. These data thus suggest that the management of salt intake and inhibition of RAAS activity is required to suppress the development of salt-sensitive hypertension in post-AKI patients.

## Figures and Tables

**Figure 1 ijms-23-07831-f001:**
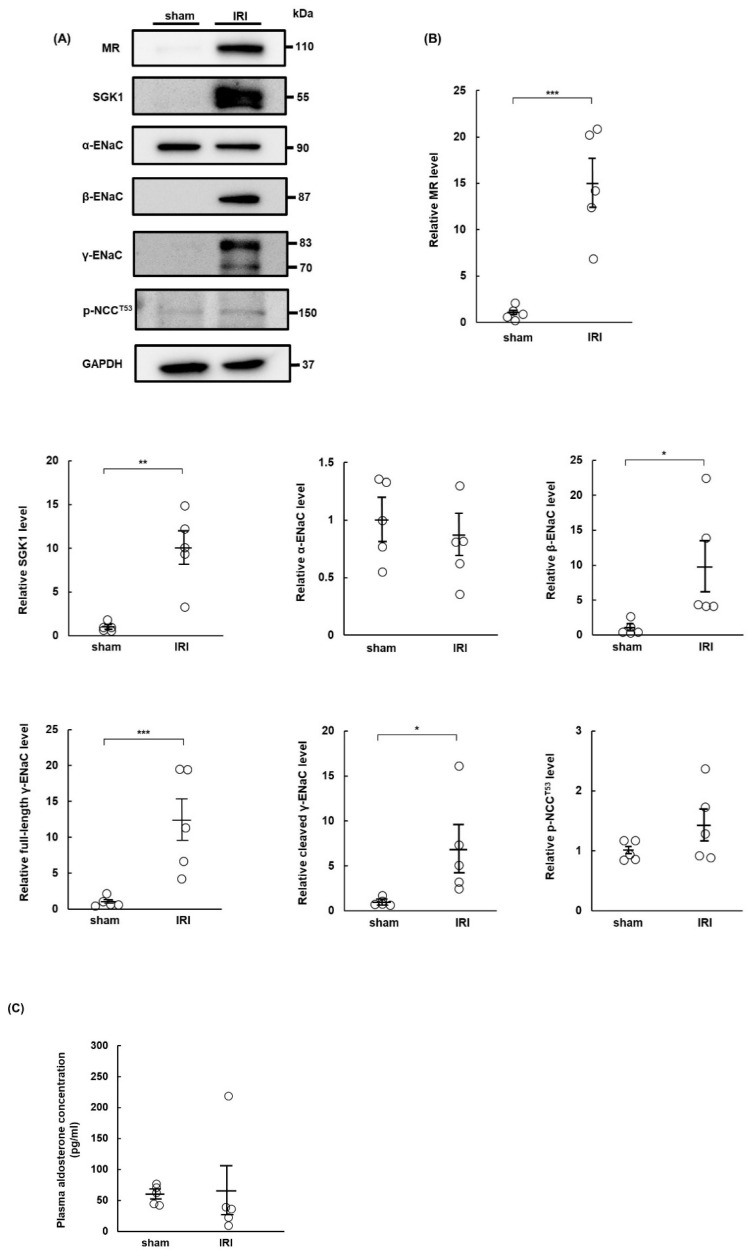
Expression of mineralocorticoid receptor increases in rats 7 days after ischemia–reperfusion injury. Sprague Dawley rats underwent sham or ischemia–reperfusion injury (IRI) surgeries 14 days after right nephrectomy. IRI was induced by clamping the left renal artery. The rats were euthanized on day 7 after the operations, and kidney tissues were collected: (**A**) Western blots show expression levels of the mineralocorticoid receptor (MR), serum and glucocorticoid-regulated kinase 1 (SGK1), α-epithelial sodium channel (α-ENaC), β-ENaC, full-length γ-ENaC, cleaved γ-ENaC, phospho-NaCl cotransporter channel (Thr53) (p-NCC^T53^), and glyceraldehyde 3-phosphate dehydrogenase (GAPDH) in kidneys from sham- and IRI-surgery rats. GAPDH was used as a loading control; (**B**) relative protein levels of MR, SGK1, α-ENaC, β-ENaC, full-length γ-ENaC, cleaved γ-ENaC, and p-NCC^T53^ in the sham and IRI groups. Band intensities were normalized to GAPDH; (**C**) plasma aldosterone concentrations in the sham and IRI groups. Values are presented as mean ± standard error (n = 5 rats per group). Data were analyzed by Student’s *t*-test for comparisons between two groups. * *p* < 0.05, ** *p* < 0.01, *** *p* < 0.001.

**Figure 2 ijms-23-07831-f002:**
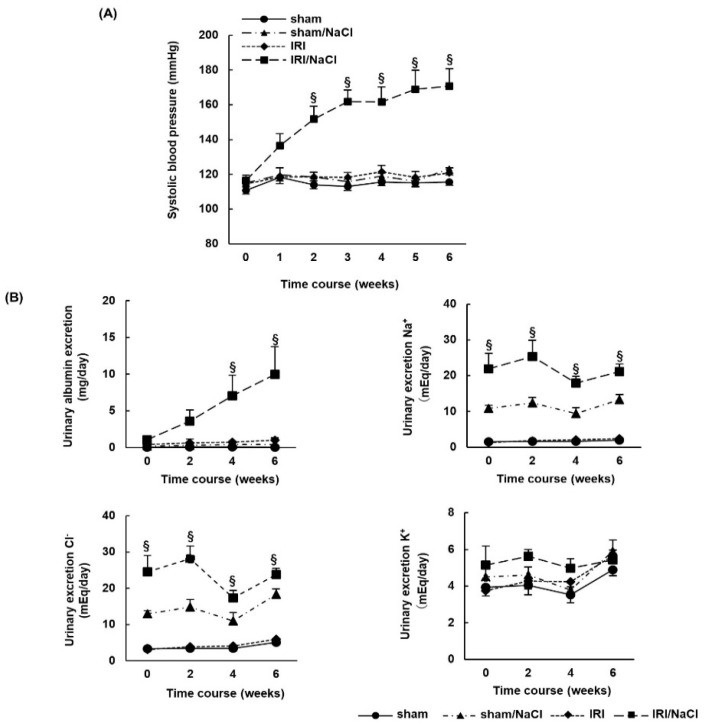
Drinking water containing 1.0% NaCl causes hypertension and renal damage in rats with ischemia–reperfusion injury. Sprague Dawley rats underwent sham or ischemia–reperfusion injury (IRI) operations and were provided with regular drinking water or 1.0% NaCl solution for 6 weeks: (**A**) systolic blood pressure was measured once per week during the observation period; (**B**) urinary excretion of albumin, Na^+^, K^+^, and Cl^−^; (**C**) representative images of hematoxylin–eosin (HE) staining, Masson’s trichrome (MT) staining, and immunohistochemical staining for collagen type 3 (Col-III) showed typical morphological changes in renal tissue (scale bar = 100 µm). Cell numbers (HE staining), % fibrotic area (MT staining), and Col-III positive areas were quantified for each group from stained images and presented in the graphs; (**D**) representative Western blots show expression levels of alpha-smooth muscle actin (α-SMA) and glyceraldehyde 3-phosphate dehydrogenase (GAPDH) in renal tissue from each experimental group. GAPDH was used as a loading control. Graph shows relative protein expression levels of α-SMA. Data are presented as mean ± standard error (n = 6 rats per group). Data were analyzed by one-way analysis of variance, followed by Tukey’s test. § *p* < 0.05 for comparisons of blood pressure and urinary data; * *p* < 0.05, ** *p* < 0.01, and *** *p* < 0.001 for histological quantification and relative α-SMA levels.

**Figure 3 ijms-23-07831-f003:**
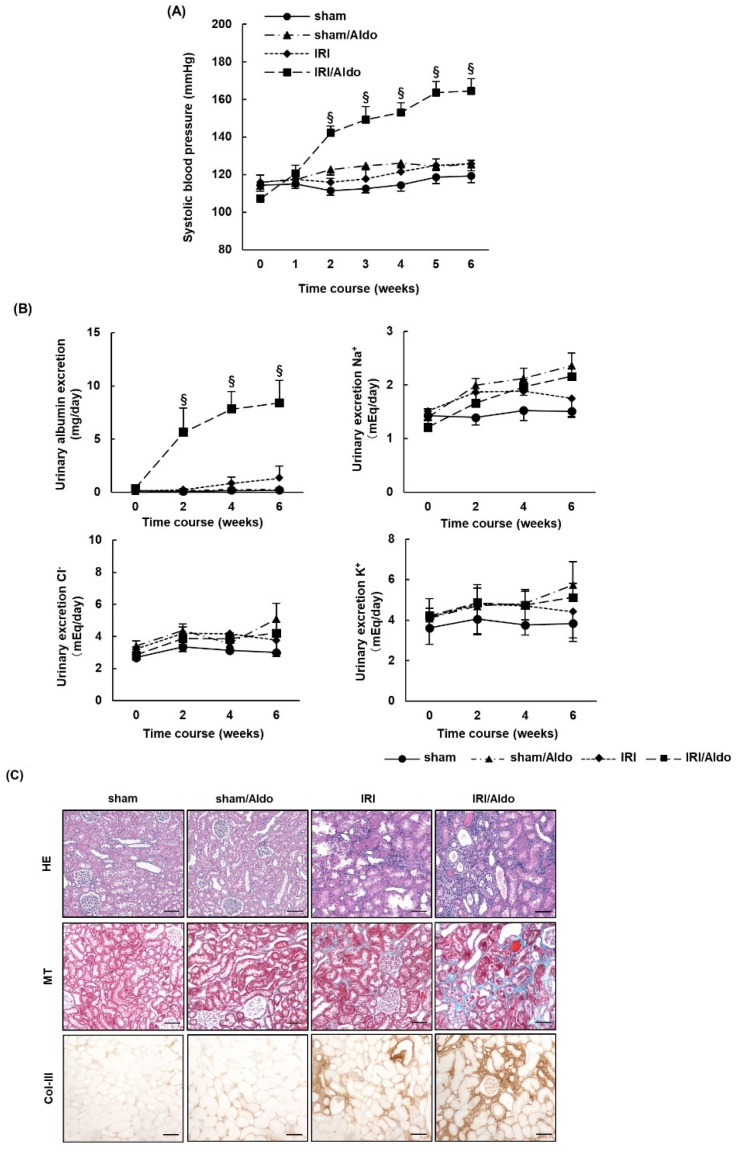
Infusion of aldosterone induces hypertension and renal damage in rats with ischemia–reperfusion injury. Sprague Dawley rats underwent sham or ischemia–reperfusion injury (IRI) surgeries and were infused with vehicle or aldosterone for 6 weeks: (**A**) systolic blood pressure was measured once per week during the observation period; (**B**) urinary excretion of albumin, Na^+^, K^+^, and Cl^−^; (**C**) representative images of hematoxylin and eosin (HE), Masson’s trichrome (MT), and immunohistochemical staining for collagen type 3 (Col-III) showed typical morphological changes in renal tissue (scale bar = 100 µm). Graphs show quantification of cells using HE staining, fibrotic areas using MT staining, and % Col-III positive area; (**D**) representative Western blots of renal tissue from each group show expression levels of alpha-smooth muscle actin (α-SMA) and glyceraldehyde 3-phosphate dehydrogenase (GAPDH). GAPDH was used as a loading control. Graph shows relative protein expression of α-SMA. Data are presented as mean ± standard error (n = 6 rats per group). Data were analyzed by one-way analysis of variance followed by Tukey’s test. § *p* < 0.05 for differences between groups for blood pressure and urinary data; * *p* < 0.05, ** *p* < 0.01, and *** *p* < 0.001 for remaining data.

**Figure 4 ijms-23-07831-f004:**
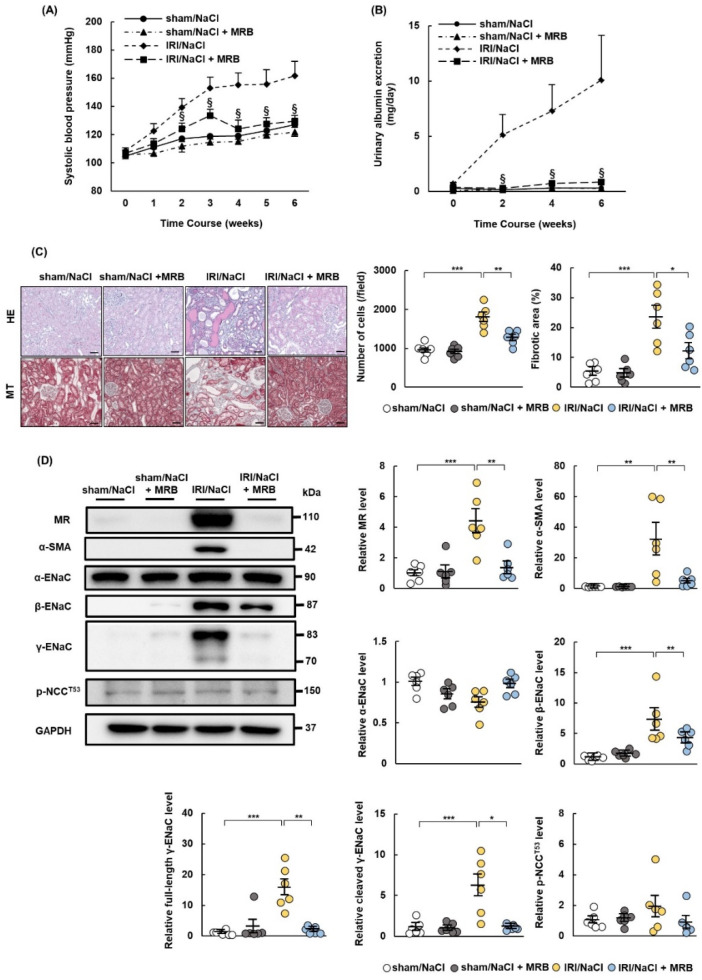
Esaxerenone ameliorates hypertension and renal damage in rats with ischemia–reperfusion injury and salt overload. Sprague Dawley rats underwent sham or ischemia–reperfusion injury (IRI) surgeries and were provided with 1.0% NaCl solution and administered esaxerenone, a non-steroidal mineralocorticoid receptor (MR) blocker, or vehicle orally for 6 weeks: (**A**) systolic blood pressure was measured once per week during the observation period; (**B**) urinary excretion of albumin; (**C**) representative images of hematoxylin–eosin (HE) and Masson’s trichrome (MT) staining showed typical morphological changes in renal tissue (scale bar = 100 µm). Graphs show quantification of cell numbers and fibrotic areas in HE- and MT-stained images, respectively; (**D**) representative Western blots show expression levels of MR; alpha-smooth muscle actin (α-SMA); α, β, and γ subunits of the epithelial sodium channel (ENaC); phospho-NaCl cotransporter channel (Th53) (p-NCC^T53^); and glyceraldehyde 3-phosphate dehydrogenase (GAPDH). GAPDH was used as a loading control. Graphs indicate relative expression levels of each protein. Values represent mean ± standard error (n = 6 rats per group). Data were analyzed by one-way analysis of variance followed by Tukey’s test. § *p* < 0.05 for blood pressure and urinary albumin; * *p* < 0.05, ** *p* < 0.01, and *** *p* < 0.001 for remaining data.

**Figure 5 ijms-23-07831-f005:**
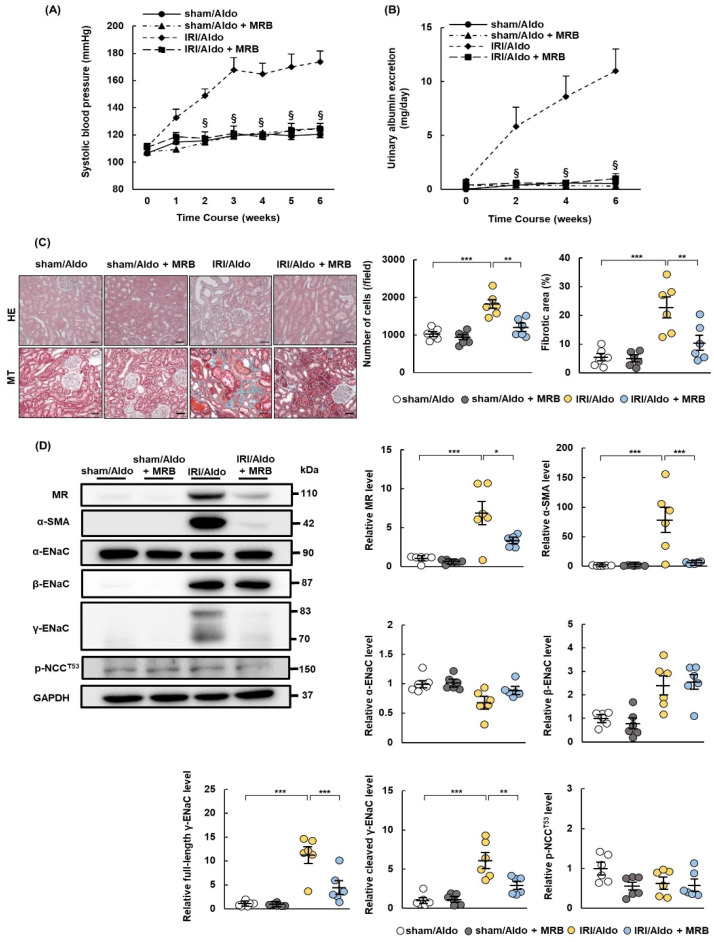
Esaxerenone ameliorates hypertension and renal damage in rats with ischemia–reperfusion injury and aldosterone infusion. Sprague Dawley rats underwent sham or ischemia–reperfusion injury (IRI) operations and were infused with aldosterone. Esaxerenone, a non-steroidal mineralocorticoid receptor (MR) blocker, or vehicle was administered to sham or IRI rats for 6 weeks: (**A**) systolic blood pressure was measured once per week during the observation period; (**B**) urinary excretion of albumin; (**C**) representative images of hematoxylin and eosin (HE) and Masson’s trichrome (MT) staining showing typical morphological changes in renal tissue (scale bar = 100 µm). Graphs show quantification of cell numbers using HE staining and fibrotic area using MT-stained images; (**D**) representative Western blots show expression levels of MR; alpha-smooth muscle actin (α-SMA); α, β, and γ subunits of the epithelial sodium channel (ENaC); phospho-NaCl cotransporter channel (Th53) (p-NCC^T53^); and glyceraldehyde 3-phosphate dehydrogenase (GAPDH) in renal tissue. GAPDH was used as a loading control. Graphs show relative expression levels of each protein. Data are presented as mean ± standard error (n = 6 rats per group). Data were analyzed by one-way analysis of variance, followed by Tukey’s test. § *p* < 0.05 for blood pressure and urinary data; * *p* < 0.05, ** *p* < 0.01, and *** *p* < 0.001 for all other data.

**Figure 6 ijms-23-07831-f006:**
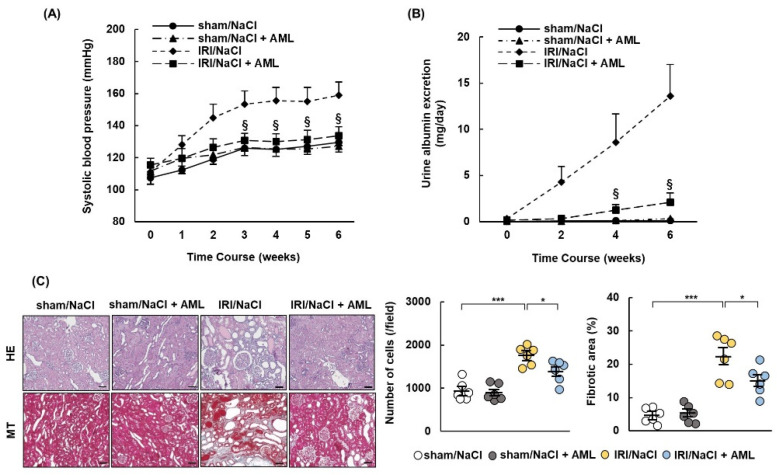
Amiloride ameliorates hypertension and renal damage in rats with ischemia–reperfusion injury and salt overload. Sprague Dawley rats underwent sham or ischemia–reperfusion injury (IRI) surgeries and were then provided with 1.0% NaCl solution. Amiloride (AML) or vehicle was administered to sham and IRI rats for 6 weeks: (**A**) systolic blood pressure was measured once per week during the observation period; (**B**) urinary excretion of albumin; (**C**) representative images of hematoxylin–eosin (HE) and Masson’s trichrome (MT) staining show typical morphological changes in renal tissue (scale bar = 100 µm). Graphs show quantification of cells and fibrotic areas using HE- and MT-stained images, respectively; (**D**) representative Western blots show expression levels of alpha-smooth muscle actin (α-SMA); α, β, and γ subunits of the epithelial sodium channel (ENaC); mineralocorticoid receptor (MR); and glyceraldehyde 3-phosphate dehydrogenase (GAPDH). GAPDH was used as a loading control. Graphs show relative expression levels of each protein. Data are presented as mean ± standard error (n = 6 rats per group). Data were analyzed by one-way analysis of variance, followed by Tukey’s test. § *p* < 0.05 for blood pressure and urinary data; * *p* < 0.05, ** *p* < 0.01, and *** *p* < 0.001 for remaining data.

**Figure 7 ijms-23-07831-f007:**
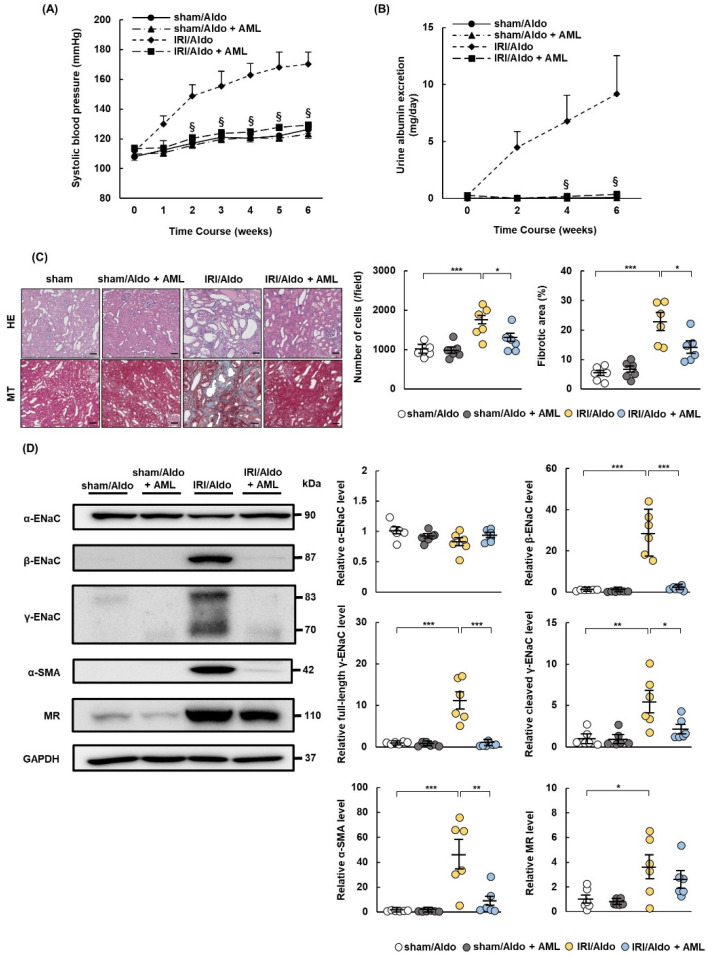
Amiloride ameliorates hypertension and renal damage in rats with ischemia–reperfusion injury and aldosterone infusion. Sprague Dawley rats underwent sham or ischemia–reperfusion injury (IRI) operations and were infused with aldosterone. Amiloride (AML) or vehicle was administrated to sham or IRI rats for 6 weeks: (**A**) systolic blood pressure was measured once per week during the observation period; (**B**) urinary excretion of albumin; (**C**) representative images of hematoxylin and eosin (HE) and Masson’s trichrome (MT) staining show typical morphological changes in renal tissue (scale bar = 100 µm). Graphs show quantification of cell number and fibrotic area using HE- and MT-stained images, respectively; (**D**) representative Western blots show expression levels of alpha-smooth muscle actin (α-SMA); α, β, and γ subunits of epithelial sodium channel (ENaC); mineralocorticoid receptor (MR); and glyceraldehyde 3-phosphate dehydrogenase (GAPDH). GAPDH was used as a loading control. Graphs show relative expression levels for each protein. Values represent mean ± standard error (n = 6 rats per group). Data were analyzed by one-way analysis of variance, followed by Tukey’s test. § *p* < 0.05 for blood pressure and urinary data; * *p* < 0.05, ** *p* < 0.01, and *** *p* < 0.001 for the remaining data.

## Data Availability

The data analyzed during the present study are available from the corresponding author upon reasonable request.

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
