# Peer review of "Upregulation of Mineralocorticoid Receptor Contributes to Development of Salt-Sensitive Hypertension after Ischemia–Reperfusion Injury in Rats"

_ijms, 2022, doi:10.3390/ijms23147831_

Round 1
Reviewer 1 Report
In the paper entitled « Upregulation of mineralocorticoid receptor contributes to development of salt-sensitive hypertension after ischemic reperfusion injury in rats» Matsumoto and collaborators provide evidences indicating that acute kidney injury induced by ischemic reperfusion injury is associated with a strong upregulation of the mineralocorticoid receptor (MR) in the kidney. This is associated with the up-regulation of downstream MR targets such as the epithelial Na channel ENAC (mainly beta/gamma subunits) and SGK1. Upon challenges with either high salt (1% NaCl drinking water) or chronic aldosterone infusion, systolic hypertension develops associated to renal injury with fibrosis and inflammation. Esaxerenone, a non-steroidal MR antagonists blunted these changes. Of note amiloride, a ENAC blocker, had similar effects.
The study is interesting and well done with appropriate control experiments. Nevertheless several concerns need to be addressed to support the conclusion
1) The upregulation of MR, at the protein level, detected by western blot is very strong and somewhat surprising. It is noteworthy that anti-MR antibodies are quite difficult to use and that non-specific staining could be observed. As this is a major finding supporting the hypothesis and conclusions, this should be confirmed by a second unrelated anti-MR antibody
2) Immunostaining of MR should be provided. MR is mainly expressed in the ASDN in physiological setting. Such strong upregulation by IRI-AKI should be better analyzed and may reflect additional expression outside of the ASDN (inflammatory cells? fibroblasts? mesangial cells/podocytes) which is quite relevant in this context
3) The processing of ENaC is important during ENac activation, as mentioned by the authors, especially by aldo/MR. While such activation/cleavage is exemplified in the study for the gamma subunit, it is surprising that this does not occur for the alpha subunit (a 30 KD band should be visible, using appropriate antibodies).
4) Plasma aldo levels in IRI seems rather decreased (only one sample is very high and drives probably the analysis toward no statistical difference). How is aldo level in the initial series with 6 rats shown in the sup data figS1?
5) MR upregulation is fully blunted by esaxerenone, while amiloride, despite improving all parameters, has no effect. What is the proposed mechanism? along the same line, what is the proposed mechanism for upregulation of the MR induced upon IRI-AKI ?
Author Response
We are grateful to Reviewer 1 for their critical comments and useful suggestions, which have helped us to improve our manuscript considerably.
Please see the attachment.

Reviewer 2 Report
Review ijms-1786536
Matsumoto et al., “Upregulation of mineralocorticoid receptor contributes to development of salt-sensitive hypertension after ischemic reperfusion injury in rats”.
The authors aim to study the role of the mineralocorticoid receptor in the progression of IRI-induced salt-sensitive hypertension in rats. Following induction of IRI, rats were provided normal water or water with 1% NaCl or infused with aldosterone or its vehicle.
Esaxerenone, a non-steroidal MR blocker attenuated blood pressure and urinary protein excretion in IRI/NaCl and IRI/Aldo rats. Additionally, gENaC protein abundance and renal damage was attenuated. The authors concluded that IRI-induced upregulation of the MR in the distal nephron contribute to increased expression and thus activity of ENaC and NCC, allowing salt overload or activation of the RAAS to cause hypertension.
Although their data are of importance in the field, the finding is not completely new, since spironolactone, a mineralocorticoid receptor blocker was already shown in 2007 to prevent renal ischemia-reperfusion injury (Mejia-Vilet et al., 2007), and thus prevented the development of chronic kidney disease (CKD) after a severe (45 min.) ischemic injury (Int J Biol. Sci 2015, 2019), proposing MR antagonism as a promising therapeutic approach to prevent IRI.
While the outcome using a MR blocker is thus confirmed, the presentation and interpretation of the single data is less clear and needs ameliorations. The Recent references in the field are missing and should be included in the discussion. Previous data using spironolactone should be better discussed (what are the advantages vs disadvantages of spironolactone vs esaxerenone) and included in the discussion. ENaC activity is not directly measured that should be considered for the interpretation of the data.
Specific comments:
1) The authors newly find a about 15-fold upregulation of the MR protein abundance accompanied by a 10- and 12-fold increased beta and gamma ENaC expression, respectively. This does not necessarily determine the ENaC activity. Did they also perform an amiloride or benzamil acute blockage of ENaC to determine its activity (eg. By determining the amiloride-sensitive FE of Na+)? This should be considered.
2) The authors newly applied the non-steroidal mineralocorticoid receptor blocker esaxerenone, but the dosis and the timely protocol of application is not described. This should be added in the Material and Methods section.
3) Equally, the concentration of aldosterone treatment is not indicated.
4) It is quite surprising that MR but also alpha-SMA, beta ENaC and gamma ENaC are nearly not expressed in sham-operated animals, at least it is not visible on the blots. This contradicts previous papers in the field. Any explanation?
5) Additionally, all 5-6 animals should be illustrated on the blots and the blots less cut
6) Did the authors checked endothelin-B receptor expression and endothelial nitric oxide synthase activation as described by Barrera-Chimal et al., 2019?
7) How to the authors explain (118) – no difference in plama aldosterone concentrations between IRI and IRI/NaCl rats (S2).
8) The authors should better describe what can be seen on the photos (eg., Fig. 2C- description is missing (localization), etc.?)
9) Recently, two papers discussed the activation of ENaC independent of MR signaling/activation (Wu et al., 2020; Nesterov et al., 2021). These paper should be added and discussed.
10) What is AML?
11) The authors should clearly distinguish when they discuss "hypertension" versus "salt-sensitive hypertension".
12) The discussion should be more focused on the presented data, and not presented as a review in human.
Author Response
We are grateful to Reviewer 2 for their critical comments and useful suggestions, which have helped us to improve our manuscript considerably.
Please see the attachment.

Round 2
Reviewer 2 Report
I looked through the responses to reviewers and the revised paper, but for me the requested corrections are only « cosmetically » done, and not what could had been done during a revision process (even within the 10 days of revision), namely,
Response 4 (reviewer 2) I furthermore do not agree with the authors, that for the readership »It is not important whether they (WB bands) are visible ». Then, we can omit the showing of any blots in the future.
Response 5 (reviewer 2) I requested to include all analysed Westernblot samples used in the presented blots/Figures. This is NOT DONE. The authors present some blots in the rebuttal letter, but due to variation in the band intensity, what is finally chosen to be representative. Also, the number of the lanes for each condition (now shown as blot) or then in the quantification in the manuscript do not fit (eg. Fig. 4, bENaC, condition 2, n=2 lanes are shown on the blot, but in their quantification, they state n=6 )? And there are several examples of this. This is confusing.
Response 7 (reviewer 2) I expect that the authors are considering our points raised and discuss and include their answer into the revised text. This is not done here.
Response 8 (reviewer 2) I requested a better description of the photos presented. This is not done.
Response 9 (reviewer 2) I expect as reviewer that the 2 recent papers mentioned about « MR-independent ENaC regulation » are taken into consideration here and cited and discussed. This is not done here.
Response 12 (reviewer 2): I asked that the discussion should be more focused and thus presented data should be discussed with respect to the literature. Nothing is discussed about the use of spironolactone (2007 published) and the presented data. They ignored this.